# Recent Development in Human Motion and Gait Prediction

Junwu Zhang and Monroe Kennedy III
Department of Mechanical Engineering
Stanford University, Stanford, CA 94305
Email: {junwuz, monroek} @stanford.edu

*Abstract*—**Human intent is often hard to model and predict. In fields such as robotics and biomechanics, one type of human motion that is important to model is the bipedal walking motion. As a type of legged locomotion, bipedal walking has unique advantages in daily-life environments such as stairs and muddy surfaces, compared to rolling locomotion that are used by many autonomous mobile robots. In this paper, recent development in predicting bipedal gait dynamics and the corresponding human motion trajectory is presented. Such prediction usually require two main steps: data collection and data analysis. We inspect and compare existing solutions in each of the two steps, summarize the common approaches, and discuss the potential opportunities in the field going forward.**

## I. INTRODUCTION

The modeling and prediction of human intent are crucial in many areas of robotics research. An example of such a human motion that is commonly studied is the bipedal walking motion. Bipedal walking is significant since the form of locomotion is key to mobile robots. In recent times, wheel-based robots such as autonomous vehicles and delivery robots, have seen an increase in commercial usage. Those robots are able to move efficiently on paved, smooth surfaces, however they face the "last mile" problem in the real-world environment. Realistically, our world is full of natural obstacles such as rocks, muddy surfaces, and complex terrains such as stairs and hills. These environments are hard for wheeled locomotion and more suitable for legged mechanisms [48].

Bipedal gait analysis and motion prediction can be used in many robotics applications. Exoskeletons and active prosthesis could use gait data to help train patients in rehabilitation [2], gait monitoring and prediction could help with fall prevention [4], gait analysis can be used in bipedal robots control [9], and gait cycle could help predicting pedestrian intent at intersections [16].

There is an abundance of literature on human motion data collection, modeling and prediction in various applications. In this work we select the ones that are most relevant to human intent prediction in the context of bipedal walking. Recent papers were selected to reflect the state of the art. The citations and impact factors of the references are also taken into account, especially for more established fields such as data collection.

The rest of the paper is structured as follows. Data collection will be covered in Section II, where various methods for capturing bipedal gait data will be reviewed. Data analysis will be covered in Section III, which includes mostly learning-based methods but also some traditional approaches. Discussions on open research questions and opportunities will be presented in Section IV, with the conclusion presented in Section V.

## II. DATA COLLECTION

To study bipedal gait, collecting gait-related data is the first necessary step. Various methods exist, but overall, data collection can be done with two primary considerations: human and environment observations.

Data collection using human observations tracks the movements in the human body itself. Tools such as Kinect, regular stereo cameras and motion capturing system can be used to log visual data of human motion. On the other hand, the surrounding obstacles and terrain are also helpful to infer gait information, and tools such as wearable cameras and reaction force plates can help collect those data.

Worth noting is that most data collection system are coupled with some format of data processing. The focus in this section is the data collection setup itself and how they might contribute to the subsequent data analysis.

### A. Kinect Sensors

Most visual data collection methods for bipedal walking are done using one or multiple camera from the perspective of the environment. This type of approach is popular due to its relative simplicity, where cameras can easily see the entirety or part of the human walking motion. One commonly used visual sensor is Kinect, a Microsoft product that accompany their Xbox 360 gaming system. It is equipped with a depth camera based on infrared (IR) sensing that could perform skeletal tracking and gesture recognition [42].

Preis et al. [32] have built a system based on the Kinect with the goal of gait recognition. Specifically, they collect 13 biometric features including the height, the length of limbs, and the step length directly using the skeleton points generated by the Kinect software development kit (SDK). Afterwards, the actual pose estimation is done based on a view-invariant approach [3] that includes gait cycle, stride and height estimations. Using this setup, a $85.1\%$ feature classification success rate is achieved when using a Naive Bayes classifier.

Staranowicz et al. [39] have presented a multi-Kinect framework. The proposed system estimates the extrinsic and

| Ref | Objective | Method | Pros | Cons |
|---|---|---|---|---|
| [32] | Gait recognition from visual data | Used Microsoft Kinect as visual sensor, direct skeletal data points collection from Kinect software development kit (SDK) | Relatively simple to use and set up, achieves decent recognition accuracy | Experiment setup could be expanded to track more parameters in multi-human scenarios |
| [39] | Gait monitoring for fall prevention with easy-to-use Kinect-based system | Two Kinect sensors are used, intrinsic and extrinsic calibration parameters are estimated, focuses on ease-of-use and accuracy | Easy to set up, multiple Kinects provide additional volume and accuracy over single-Kinect setups | The two-camera setup is deterministic, a more generalizable calibration algorithm could see more applications |
| [43] | Human gait and pose collection, recognition and tracking | Single Kinect hidden for unobtrusive data capturing, raw depth streaming from Kinect sensor is used instead of the skeletal joints tracking provided by Kinect SDK | Able to recognize human gait for a longer range than native Kinect SDK supports | Single Kinect might not be enough to capture all areas of a room for fall prevention |
| [46] | Human gait estimation from visual data | Used a camera mounted on a person's thigh, close to the waist area | Thigh-mounted camera is mobile, can be used outdoors and can capture environmental information around a person | Could use more testing to find the best point and/or orientation to mount the camera, also could adjust the type of camera mounted |
| [36] | Gait data collection (joint angle measurements) | Strategically placed inertial measurement units (IMUs) on thigh, calf and foot | The proposed method is agnostic to sensor placements and initial human postures | Joint angle measurements is still only 2D for now, plug-and-play wireless data communication is not yet supported |
| [47] | Gait initiation estimation and detection | Electromyography (EMG) sensors for gait initiation estimation, and IMUs for gait initial movement detection | Demonstrates that EMG could be used to predict initial movements in gait motion | EMG prediction only seems useful when prosthetic leg leads in the gait motion, EMG sensors are harder to work with and requires a much higher sampling frequency |
| [31] | Gait data collection and biomechanics analysis | Proposed 3D GAIT, a 3D biomechanical gait data collection system where Vicon markers are placed on various locations of human subject | Automated system, relatively easy to use among marker-based methods, designed for best practices in biomechanics using big data | Marker system are still relatively more complicated to deploy than other alternatives |
| [27] | Gaze and fully-body gait kinematics data collection (for studying relationship between gaze and footstep planning) | Mobile eye tracker for gaze tracking, IMU-based system for motion and gait capturing | Captures both gaze and gait data accurately and completely in a mobile package, sensors can be obtained off-the-shelf for easier experiment set up | Though the system works in an outdoor environment, there are still a high number of sensors, making the system complicated in a mobile scenario |

intrinsic calibration parameters for each individual Kinect, as well as the rigid-body transformation between multiple Kinects. One of the novel contributions of the paper is the multi-Kinect calibration. A set of 3-D sphere centers $\left\{^{D_j}\mathbf{X}_f\right\}_{f=1}^{F}, \left\{^{D_i}\mathbf{X}_f\right\}_{f=1}^{F}$ is collected, where $D_i$ and $D_j$ represents depth frame of either Kinect, and $F \geq 3$ is the number of frames. An initial estimate of $^{D_j}_{D_i}\mathbf{H}$ can then be obtained using a 3-step process. Afterwards, bundle adjustment is done to refine the set of initial estimates.

Kinect can also be used for fall detection and prevention. Stone and Skubic [43] have presented a system where the Kinect is placed above the front door of an elderly residence. The computer processing the data of the Kinect is hidden in a cabinet to make the setup not easy to notice for the residents. This method uses the raw depth streaming from the Kinect rather than the skeleton points provided by the SDK like in the previous study, due to the range limitations of the SDK-provided skeleton tracking. In this setup, foreground and background pixels are differentiated based on a mixture of distributions approach [40], projected onto a Kinect-based 3-D space, before being translated into world-based coordinates. This setup, paired with a two-stage fall detection algorithm, can successfully detect 98% of standing falls in an indoor elderly residence study.

### B. Wearable Camera

Though not the most widely used method for data collection due to its relative complexity, egocentric visual data can be helpful in many cases. Such data are useful especially in dynamic and outdoor environments where static external cameras are hard to position and mobile wearable sensors are desired. Watanabe et al. [46] have shown a unique setup where camera was mounted on the human thigh. The camera is mounted facing downward as it is sufficient to capture environmental data and can reduce accumulated sensor errors, but forward-facing cameras can contain more information. Pose estimation from this single camera is then performed with algorithm based on extended Kalman filter (EKF).

Cameras can be head-mounted to provide first-person point-of-view (POV) visual data. These data are usually available as datasets open for the public for further training, learning and benchmarking [38, 37].

Overall, we observe that visual data can be captured by either Kinect-based systems or regular stereo cameras. Kinect sensors that perform depth sensing using active IR sensors are more suitable in indoor environments due to less interference by sunlight, and can be effective no matter if the scene is visual-feature rich. On the other hand, stereo cameras can work in either indoor or outdoor scenarios, but accuracy

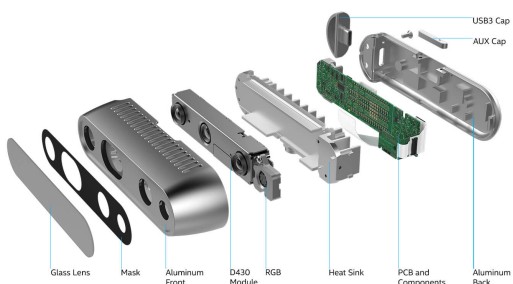

Fig. 1. Intel® RealSense™camera [1]

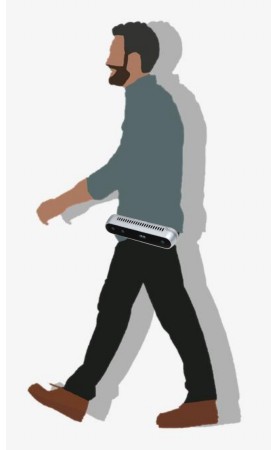

Fig. 2. A possible approach for waist-mounted wearable RealSense™camera, inspired by [46]

depends on sufficient number of features in the scene. Combination of Kinect and stereo camera can be used to reduce environmental limitations, with an example being the Intel® RealSense™camera [1] shown in Fig. 1. A possible usage of RealSense™inspired by [46] is shown in Fig. 2.

### C. Wearable Inertial Sensors

Using sensors such as IMUs is another approach to gather gait kinematics data. Seel et al. [36] have presented an IMU-based system for joint angle calculation in gait motion. This method makes IMU measurements agnostic of factors such as sensor placements and initial human postures. Knee joints are modeled as mechanical hinge joints, where kinematic constraints are used to obtain position vectors and direction vectors of knee muscles axis in sensor coordinates. IMUs are placed on the thigh, calf and foot of the human subjects. Experiments are carried out on transfemoral amputeess (TFAs) and results are compared with a baseline gathered by motion capture system, showing a root-mean-square error (RMSE) of about 1 deg on ankle joints measurements.

Li et al. [23] have shown a similar but more mobile system composed of multiple IMUs on hip, knees and ankles, with force sensors built into human subject's shoes. The collected joint angle data also matched that of an optical motion capturing system, proving its potential applications in ambulatory and mobile environments.

### D. Wearable Biosensors

Wentink et al. [47] have looked into whether it is feasible to use EMG for gait initiation detection and prediction, with the goal of helping active prosthesis development. The study includes TFA as human subjects, and investigates two scenarios in the walking motion: prosthetic leg leads or intact leg leads. Self-adhesive EMG sensing electrodes are placed on eight upper leg muscles of the residual part of the prosthetic leg. 16 bipolar channel Porti-system is used to measure EMG at 2048 Hz. During experiments, subjects are asked to stand in upright positions initially, walk 5 steps, turn around and walk back to the original point, with pauses in between for posture measurements. EMG data are high pass filtered at 10 Hz and low pass filtered at 500 Hz with a second order Butterworth filter. The results show that EMG sensors can predict initial movement up to 138ms in advance, and are most effective when the prosthetic leg was leading in the walking motion.

### E. Motion Capturing

Wearable markers, and most famously a marker system called Vicon, is a common optical gait analysis tool often used in clinical applications [28, 14]. Gait data collection using both Kinect and Vicon is compared by Pfister et al. [30], and it is found that Kinect and Vicon system correlate well in stride timing measurements, but not well enough in hip and knee measurements.

Phinyomark et al. [31] have presented a 3D GAIT system for gait data collection. This is a motion capture system for treadmills, so free environment might not be supported. This also means that sensors are not mounted on the humans directly. Only the reflective markers for motion capture are put on human subjects.

Matthis et al. [27] have showcased a data collection system for studying the relationship between gaze and foothold planning. A Positive Science mobile eye tracker is used to record gaze data. The eye tracking also works in sunlight, with the help of an infrared-blocking face shield. For gait data, a full body motion capture system from Motion Shadow is used. Since the system can be strapped to the human tester, a traditional indoor environment with external motion capture sensors are not needed. Finally, a backpack-mounted MacBook Air is worn by the tester for easy data collection in outdoor environments. This suite of sensors helps researchers identify the consistent importance of eye gaze to the planning of footsteps, especially on relatively complex terrains.

### F. Additional Methods

Quite a few other methods for gait measurements are possible. Real life applications such as security cameras often capture various views of gait. Chattopadhyay et al. [6] have demonstrated gait recognition from frontal view, and Zhao et al. [49] have shown that multiple cameras can be used to capture 3D gait information. Visual-inertial odometry (VIO) technologies [33, 22] are also potential solutions to gait analysis if they are able to be used as wearable sensors.

In addition, traditional external cameras can be used with novel computer vision (CV) algorithms to capture motion. Joo et al. [19] have presented a method that allows markerless motion capture for both full body motion and lower-level details such as facial expressions. Existing models for face, hand and body motion are combined in one skeleton hierarchy. A new model is then derived from that hierarchy to capture more total body motion information with less parameters.

## III. DATA ANALYSIS

Data processing is the actual step where the analysis and prediction of gait and the resulting motion is generated. In some cases, predictions are focused on parameters used to characterize gait, including gait cycle frequency, joint positions, joint kinematics and so on. In other scenarios, the higher-level human trajectory is the objective of the prediction. Most prediction methods in recent research are based on deep learning (DL), with optimization or state-estimation methods also possible. A brief categorization of DL methods reviewed in this paper in shown in Fig. 3.

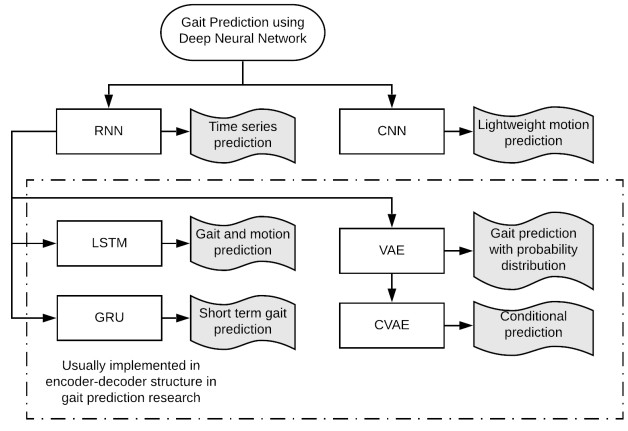

Fig. 3. Popular deep neural network (DNN) methods used in gait prediction

### A. *Sequence-to-sequence (seq2seq) Recurrent Neural Network (RNN)*

RNN has been commonly used for learning sequential data [10, 24], and seq2seq [44] is a type of problem where RNN can be utilized to train models to solve machine translation problem. Martinez et al. [26] have proposed a seq2seq architecture using RNN for smooth and light weight short-term human motion prediction. A single gated recurrent unit (GRU) is used rather than the more popular multi-layer Long Short-Term Memory (LSTM), where ground truth is fed into an encoder network with the error computed on a decoder network. Parameter tuning is not necessary as decoder took in its own samples as input. A residual connection is added between the RNN input and output to model velocity better, since continuous motion might better be predicted based upon velocity instead of pose configuration. The results are optimal

where zero-velocity motion prediction is much more continuous and smooth compared to that of traditional methods.

### B. *LSTM*

LSTM is a type of RNN architecture that aims to tackle the vanishing gradient problem [15], thus having better performance in predicting time-series data.

Du et al. [13] have shown a LSTM-based method using biomechanics-inspired loss function for pedestrian pose and gait prediction. More specifically, the goal of the prediction is full-body 3D mesh represented in Skinned Multi-Person Linear (SMPL) [25] parameters. The basic network architecture includes a two-layer stacked LSTM RNN followed by a fully connected (FC) neural networks (NN) layer, with each LSTM layer consists of 32 units. The loss function is uniquely designed according to biomechanics and can be written as:

$$\min L = L_c + \lambda_1 L_s + \lambda_2 L_g \tag{1}$$

where $L_c$ is gait cycle loss, $L_s$ is body mirror symmetry loss, $L_g$ is the loss based on volume from the ground plane and $\lambda_1, \lambda_2$ are regularization parameters. Experimentation on the PedX dataset [21] returns more accurate and biologically realistic result while being more robust to noise.

### C. *Seq2seq based on GRU with Attention Mechanism*

Sang et al. [35] have proposed a RNN model consists of an encoder part based on GRU [8] and a decoder part with attention mechanism. When decoding, the decoder makes the input of each moment different according to time. This is helpful to better learn the correlations between multiple content modalities. In this framework, the output is no longer a fixed-length vector, but rather a vector containing multiple subsets for selective decoding. The attention distribution $\alpha_{tj}$ is calculated from the degree of correlation $e_{tj}$ among encoder output vectors with a softmax normalization operation. This framework shows accuracy in motion prediction in a range around 4 s, much longer than the referenced methods.

### D. *Feature Learning*

Guo and Choi [17] have divided the prediction into long-term and short-term tasks, and proposed a prediction model that learns using local feature representations. This improves upon previous methods based on feature learning[5], where body dynamics are partitioned into local features to account for different moving dynamics for different parts of the body. For long-term prediction, a network called SkelNet is proposed. Standard feed-forward network is used as the basic sequence generator and residual connection used to predict velocity instead of pose. Leaky Rectified Linear Units (LReLUs), dropouts are added to reduce overfitting, and the first few layers of the network are split into five branches according to five different sets of human body components. For shorter term predictions, a single-GRU-based RNN network is trained alongside SkelNet. The output of both networks is then fed into another feed-forward NN for merging, with the new resulting system called Skel-TNet. Ablation study reveals the

two networks show decent performance while only needing 1/30 number of parameters compared to other state-of-the-art methods.

### E. Context-aware Prediction

Since human motions are often influenced by interactions with other agents, Corona et al. [11] have presented a context-aware architecture using semantic-graph model and RNN. The graph model parameterizes agents as nodes and interaction as edges, where the interactions are learned through a graph attention layer. The learned graph model is fed into a RNN with two branches for prediction. In one branch, baseline human motion prediction is generated using basic encoder-decoder RNN with residual layer. In another branch, interactions and context feature vectors are predicted to be used together with the baseline to generate context-aware human motion prediction. Experimental results show that this approach performs much better than context-less methods for both human and object predictions.

### F. Conditional variational autoencoder (CVAE)

Many human motion prediction algorithms generate deterministic outputs, but uncertainties are common in the prediction of time series such as trajectories, and are sometimes helpful. Ivanovic and Pavone [18] have proposed Trajectron, a deep generative model that generates a distribution of trajectory prediction for multimodal and multi-agent scenarios. The model combines aspects of CVAE (an extension of variational autoencoder (VAE)), LSTM and spatiotemporal graphical models. The multi-agent relations are captured in a graph model, where the node history and future are encoded using 32-unit LSTM and edge influences are encoded using 8-unit LSTM with attention mechanism. The encoder output are concatenated and latent variable $z$ is sampled incorporating a CVAE structure. The 128-unit LSTM decoder then outputs a Gaussian Mixture Model (GMM) prediction for trajectory sampling. The experimental results show more accurate prediction and much fast trajectory generation compared to existing methods.

### G. Convolutional neural networks (CNN)

Nikhil and Tran Morris [29] have presented an end-to-end CNN approach for motion prediction. This differs from traditional LSTM-based methods, as the authors argued that temporal motion prediction is inherently continuous, and the spatial and temporal correlations might be exploited better by CNN. The trajectory history is directly fed into the model, where they are padded to a fixed length using a FC layer and passed into stacked convolutional layers. The output of the last layer is then concatenated and fed into another FC layer, which generates all predicted positions across the prediction horizon at once. The layers can be easily parallelized. Ablation study shows that this method predicts more accurately than those that predict sequentially by time steps. This is most likely due to the reduced error propagation when predicting all time steps at once.

### H. Additional Methods

Soo Park et al. [38] have shown that motion prediction is also possible if we just have egocentric stereo images. A 2.5D representation called EgoRetinal map is constructed to allow motion prediction using trajectory-optimization incorporating both visual and spatial data. Cheng et al. [7] have proposed a

TABLE II
COMPARISON BETWEEN DATA ANALYSIS METHODS

| Ref | Objective | Method | Pros | Cons |
|---|---|---|---|---|
| [13] | Predict 3D full-body meshes in future frames, given 3D poses in past frames | Biomechanics-based loss function under LSTM architecture | Novel loss function is a first step in biomechanical constraints on gait prediction, robust to noise | Independence between pedestrians is assumed, genders are not differentiated though it could be [45] |
| [35] | Human motion prediction | Seq2seq model based on GRU with attention mechanism added in the decoder | Improves the accuracy in longer-term predictions | Study was performed with single person in a single environment |
| [26] | Human motion prediction (short term) | Seq2seq with sampling-based loss function | Reduces the discontinuity at the start of the prediction, lightweight, action-agnostic | Best prediction results are dependent on high-level supervision in the form of action labels |
| [17] | Human motion prediction (long term and short term) | Learning using local structure representations, together with GRU-based RNN | First to model human pose by different body components in representation learning network, lightweight | Using feature learning on human pose prediction is not new |
| [11] | Motion prediction of human and interacting objects | Branched RNN, encoder-decoder model as baseline, incorporating interactions represented by semantic-graph model | Predicts both human and object motion well in human-environment interaction dataset | Prediction error increases with noise |
| [18] | Multi-modal multi-agent trajectory prediction (no incorporation of human pose) | Trajectron, a framework that combines CVAE, LSTM and dynamic spatiotemporal graphs | Able to generate distribution of trajectory prediction for multiple agents simultaneously, in a multi-modal environment | How robots might incorporate this for lower-level planning is not yet explored |
| [29] | Trajectory prediction (no incorporation of human pose) | Parallelizable CNN | Very lightweight model, uses CNN to leverage the continuous and temporal nature of trajectory | Model is basic with no social context or physical modeling; number of layers might be reduced by dilated convolutions |

semi-adaptable NN to account for time-varying human behaviors and bound the uncertainties in the generated predictions. Watanabe et al. [46] have used EKF-based method to estimate human gait from a wearable sensor.

## IV. DISCUSSIONS

### A. Overall Findings

The two necessary steps (data collection and data analysis) are summarized and compared in the above two sections. We found that there are relatively few variations of methods for the data collection step. Recent work on this step focuses on getting refined results, instead of having new fundamental approaches. On the other hand, data analysis has seen lots of recent research. Multiple types of machine learning methods for analysis and prediction have been proposed. More traditional or physical model-based methods for prediction such as optimization or state-estimation are less used in recent literature.

### B. Common Approaches

From literature, we can tell that certain approaches have been shown to be effective. For example, in home environments, Kinect-based system has proved to be a adequate substitute for complex motion capture system like Vicon [41], even in healthcare-related scenarios [43]. For data analysis, RNN-based methods are very popular, with various models based on LSTM, GRU and attention are proposed for specific applications. We can also see that more research start to look into the multimodal and uncertain nature of prediction, with more VAE and CVAE methods being proposed and predictions generated with a distribution.

### C. Opportunities

On the other hand, there are still opportunities for a few important aspects that researchers haven't fully addressed.

*1) High Level:* What are the use case objectives of gait prediction, and what advances are still required to achieve these objectives? Current literature have been focusing on improving the quality of gait predictions, including factors such as accuracy, smoothness and time range. But from a higher-level, the end goals of those predictions and how predictions affect those end goals are still not quite clear. For example, in-home healthcare [12] and fall prevention [41, 39] have been regularly mentioned as "potential applications" of gait predictions. However, exactly how the change in gait analysis could impact the clinical end results is still not fully discussed.

In addition, human's environment information has not been incorporated into data collection or prediction. Such information could potentially be useful, since interactions with the environment (staircases or obstacles) could influence human motion. Leveraging environmental input, gait prediction based on EMG sensors could be explored more, especially with the objective of improving lower-limb exoskeleton control. Using EMG sensors, signals from brain could be intercepted before reaching the leg to generate predictions and detect gait phases

[47, 20]. It is easy to see that the faster the human motion is (e.g. running), the shorter the effective prediction horizon from EMG would be. However, if environment information is used, prediction horizon could be extended.

*2) Opportunities in Data Collection:* As described in Section II, wearable cameras and sensors could be used as tools for gait-related data collection. This approach enjoys advantages such as being more mobile and better at capturing environment data. However, for those wearable devices, a metric for sensor limitations and invasiveness has not been proposed. For example, research shows that for some non-clinical use cases, systems based on IMUs can achieve sufficient gait capture accuracy, if benchmarked against motion-capture system. In that case, IMUs-based approaches are less invasive since a complex motion-capture space is not required. Using untethered wearable sensors also means that human could be less restricted, therefore the data captured can be more natural and realistic [34].

*3) Opportunities in Data Analysis:* In terms of predictions, one problem that could be investigated further is how might we predict for longer time horizons. Currently short term predictions are accurate within 4s [35], and CNN-based methods could be used to improve this situation due to reduced error accumulation and more continuous motion prediction. GRU and carefully-designed RNN networks that predict based on velocity could also improve long-term predictions.

Another question in feature definitions and general gait analysis is, how can "gait" be efficiently and more completely characterized? What might be the minimum number of states that are required to describe the gait, and what might those parameters be? Furthermore, what could we say about the minimum degree-of-freedom (DOF) that we need to describe gait motion? For example, assume we have a walking robot where leg motions are phase-locked and leg movements could be modeled in perfect circles, is it possible to describe this system and gait with only 1 DOF, and can the terrain be included as a parameter? If so, then a single IMU measuring ground impact might be sufficient to model this gait motion. Investigations into these questions could potentially reduce the sensors needed for gait modeling and control. Reduced gait parameters could also mean less features for learning, therefore the speed of prediction can be improved without negatively affecting the accuracy.

## V. CONCLUSION

In this work, various approaches on the data collection and analysis for human walking gait and motion is presented. Most methods are analyzed and compared in details with two tables showing their differences. Recent development in the field are summarized, and future opportunities are discussed with potential questions raised. We hope this meta-analysis could be a useful review for various methods in gait predictions, and inspire more future research.

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
