# OpenReview forum: "Recent Development in Human Motion and Gait Prediction"
_roboticsfoundation.org/RSS/2020/Workshop/RobRetro — RobRetro 2020_

### Official Review · AnonReviewer1 · 2020-06-20
**Literature review of recent papers in human motion data collection and analysis.**

**Rating:** 7
**Confidence:** 3

**Review:**

This paper presents a literature survey of recent works in human motion data collection, their relative advantages and disadvantages, and relative accuracy. it also lists analysis approaches used in literature, and the relative advantages/disadvantages of some. In general, this is a good retrospective which presents approaches that exist, and their shortcomings. It proposes some approaches that could potentially be used to mitigate some of these shortcomings. I think this paper would be useful for people starting research in human motion analysis.

Some comments:
1. I am a little unclear on the distinction between trajectory prediction, and gait prediction in this paper. All proposed approaches seems to just be predicting measured trajectories. How are these used for gait prediction? What is the definition of gait used?

2. The writing of the paper can be improved in several ways.
2a. In general the sentences are indirect and hard to read. This could be improved.
2b. With citations use plural like, Preis et al. [30] have instead of Preis et al. [30] has
2c. Section II G should be convolutional neural networks (not convoluted neural networks)


3. I have suggestions for a few more interesting papers to add to the survey:

Matthis, Jonathan Samir, Jacob L. Yates, and Mary M. Hayhoe. "Gaze and the control of foot placement when walking in natural terrain." Current Biology 28.8 (2018): 1224-1233.

Joo, Hanbyul, Tomas Simon, and Yaser Sheikh. "Total capture: A 3d deformation model for tracking faces, hands, and bodies." Proceedings of the IEEE conference on computer vision and pattern recognition. 2018.

---

### Decision · Program_Chairs · 2020-06-25

Accept